# Accuracy of Computer-Aided Dynamic Navigation Compared to Computer-Aided Static Procedure for Endodontic Access Cavities: An In Vitro Study

**DOI:** 10.3390/jcm9010129

**Published:** 2020-01-02

**Authors:** Álvaro Zubizarreta-Macho, Ana de Pedro Muñoz, Elena Riad Deglow, Rubén Agustín-Panadero, Jesús Mena Álvarez

**Affiliations:** 1Department of Endodontics, Faculty of Health Sciences, Alfonso X el Sabio University, 28691 Madrid, Spain; anuski.dp@gmail.com (A.d.P.M.); elenariaddeglow@gmail.com (E.R.D.); jmenaalvarez@gmail.com (J.M.Á.); 2Department of Stomatology, Faculty of Medicine and Dentistry, University of Valencia, 46010 Valencia, Spain; rubenagustinpanadero@gmail.com

**Keywords:** endodontics, endodontic access cavity, computer-assisted treatment, image-guided treatment, navigation system, real-time tracking

## Abstract

Purpose: To analyze the accuracy of two computer-aided navigation techniques to guide the performance of endodontic access cavities compared with the conventional access procedure. Materials and Methods: A total of 30 single-rooted anterior teeth were selected, which were randomly distributed into three study groups: Group A—guided performance of endodontic access cavities through computer-aided static navigation system (*n* = 10) (SN); Group B—guided performance of endodontic access cavities through computer-aided dynamic navigation system (*n* = 10) (DN); and Group C—manual (freehand) performance of endodontic access cavities (*n* = 10) (MN). The endodontic access cavities of the SN group were performed with a stereolithography template designed on 3D implant planning software, based on preoperative cone-beam computed tomography (CBCT) and a 3D extraoral surface scan, and endodontic access cavities of the DN group were planned and performed by the dynamic navigation system. After endodontic access cavities were performed, a second CBCT was done, and the degree of accuracy between the planned and performed endodontic access cavities was analyzed using therapeutic planning software and Student’s *t*-test. Results: Paired *t*-test revealed no statistically significant differences between SN and DN at the coronal (*p* = 0.6542), apical (*p* = 0.9144), or angular (*p* = 0.0724) level; however, statistically significant differences were observed between the two computer-aided navigation techniques and the MN group at the coronal (*p* < 0.0001), apical (*p* < 0.0001), and angular (*p* < 0.0001) level. Conclusion: Both computer-aided static and dynamic navigation procedures allowed accurate performance of endodontic access cavities.

## 1. Introduction

For appropriate treatment selection and predictable outcomes, case evaluation and precise diagnosis are the most important factors. In nonsurgical root canal treatment, the removal of tooth structure needed to prepare the access cavity can weaken the tooth up to 63% [1]. Recently, the concepts of conservative and ultraconservative “ninja” endodontic cavity preparations have emerged [2,3]. However, there are some cases in which these new approaches for freehand access cavity preparation are difficult to achieve, as in teeth with pulp canal calcification or anatomical abnormalities. In these cases, the use of splint guides based on cone-beam computed tomography (CBCT) may reduce the risk of iatrogenic complications and preserve the coronal structure of the tooth [4,5,6,7].

The development of new radiodiagnostic technologies like CBCT has led to great advances in diagnosis and treatment planning, and to the evolution of endodontics [8]. To date, designing the ideal locations for access cavities is possible through software programs that use datasets provided by CBCT scans [5,9,10]. Nevertheless, access cavities planned by therapeutic planning software require the design and manufacture of a guiding template.

Several computer-assisted methods in surgery and endodontics that try to minimize differences between preoperative treatment planning and final outcome have been described [11]. Most of them are static, since they use stereolithographic templates to perform the access cavity. However, in dental implant surgery, dynamic computer-assisted methods have been used. In these cases, an intraoperative real-time tracking device is used to monitor whether the previous treatment plan is being correctly followed [12,13,14].

Several studies have been published about dynamic computer-assisted surgery systems and their high accuracy has been proven and assessed [15,16]. It has been shown that sinus perforations or inferior alveolar nerve injuries during drilling can be reduced by using these guided systems [17].

Dynamic guidance systems for implant surgery include Robodent, Navident, X-Guide, and image-guided implantology. In endodontics, computer-aided guided navigation systems have demonstrated their potential for clinical practice in vitro [18].

The aim of this study is to analyze and compare the accuracy of two computer-aided navigation techniques to guide the performance of endodontic access cavities compared with the conventional access cavity technique, with the null hypothesis (H0) stating that there would be no difference between the accuracy of the two techniques.

## 2. Materials and Methods

### 2.1. Study Design

Thirty single-rooted anterior teeth (lower central incisors) extracted for periodontal reasons, without caries, restorations, and with clinical crown dimensions between 8.5–9.5 mm high, 4.5–5.5 mm wide M-D, and 5.5–6.5 mm wide V-L/P, were selected in this study at the Dental Centre of Innovation and Advanced Specialties at Alfonso X El Sabio University (Madrid, Spain), between January and March 2019. A randomized controlled experimental trial was conducted in accordance with the principles defined in the German Ethics Committee’s statement for the use of organic tissues in medical research (Zentrale Ethikkommission, 2003), and was approved by the Alfonso X El Sabio University Ethics Committee (Process No. 02/2019). All patients gave their informed consent to transfer the teeth for the study.

### 2.2. Experimental Procedure

The teeth were embedded into three models of epoxy resin (Ref. 20-8130-128, EpoxiCure^®^, Buehler, IL, USA) with 10 teeth each and randomly distributed (Epidat 4.1, Galicia, Spain) into the following study groups: Group A: guided performance of endodontic access cavities using computer-aided static navigation system (NemoScan^®^, Nemotec, Madrid, Spain) (*n* = 10) (SN); Group B: guided performance of endodontic access cavities using computer-aided dynamic navigation system (Navident, ClaroNav) (*n* = 10) (DN); and Group C: manual (freehand) performance of endodontic access cavities (*n* = 10) (MN).

The SN study group underwent preoperative CBCT scan (WhiteFox, Acteón Médico-Dental Ibérica S.A.U.-Satelec, Merignac, France) with the following exposure parameters: 105.0 kilovolt peak, 8.0 milliamperes, 7.20 s, and a field of view of 15 × 13 mm. Afterwards, a 3D surface scan was performed with a 3D-extraoral scanner (EVO, Ceratomic, Protechno, Girona, Spain). Datasets obtained from the digital workflow were uploaded to 3D implant planning software (NemoScan^®^, Nemotec, Madrid, Spain) to design the virtual template by matching the 3D surface scan and CBCT data, aligning the key points of the crown of the teeth. A virtual implant bur was also design using the implant planning software to create an access cavity inside each tooth with a diameter of 1.2 mm, a total length of 14 mm, and a drilling depth of 11 mm to enhance direct access to the root canal system (Figure 1A–C). After designing the virtual template (Figure 1C,D), it was fabricated by means of stereolithography technique (ProJet^®^ 6000, 3D Systems©, Rock Hill, SC, USA) (Figure 1E), except for stainless steel cylinders of 5 mm length that were manually attached to the template. Afterwards, template fitting was checked and access cavities were made to a depth of 11 mm from the incisal border. A diamond bur with a diameter of 1.2 mm on the active part, a total length of 14 mm, and a working length of 11 mm was used (Ref. 882 314 012, Komet Medical, Lemgo, Germany).

The DN study group underwent preoperative CBCT scan after the placement of a thermoplastic template (NaviStent, Navident, ClaroNav) over the dental surface of the teeth to allow a stable and retentive fixation of a radiographic marker and an attached handle with a black-and-white jaw tag fixed over the dental surface of the teeth. CBCT datasets were imported to the treatment planning software uploaded on the mounted laptop computer on the mobile unit (Navident, ClaroNav) (Figure 2A). Another black-and-white drill tag was attached to the high-speed handpiece. Both optical reference markers were calibrated and recognized by the optical triangulation tracking system comprising stereoscopic motion-tracking cameras guiding the drilling process at the planned angle, pathway, and depth of the access cavities in real time. Access cavities were monitored by the laptop computer of the computer-aided dynamic navigation system (Navident, ClaroNav) and performed with a diamond bur surface (Ref. 882 314 012, Komet Medical, Lemgo, Germany) (Figure 2B).

The MN control group underwent preoperative CBCT scan and datasets were uploaded to 3D implant planning software (NemoScan^®^, Nemotec, Madrid, Spain) to design virtual straight access cavities, but no template was used. All endodontic access cavities were performed by the same operator following the technique recommended by Clements et al. [19] and Mauger et al. [20].

Canal location was confirmed clinically in all study groups by exploring the root canal system with a #10 K-file (Dentsply Maillefer, Ballaigues, Switzerland).

### 2.3. Measurement Procedure

After performing the access cavities, postoperative CBCT scans were taken of the three groups. Virtual access cavity planning and postoperative CBCT scans of the three groups were uploaded to the 3D implant planning software (NemoScan^®^, Nemotec, Madrid, Spain) and matched to analyze the deviation angle (measured in the center of the cylinder) and horizontal deviation (measured both at the coronal entry point and the apical endpoint). Deviations were evaluated in axial, sagittal, and coronal views.

### 2.4. Statistical Tests

All variables of interest were recorded for statistical analysis with SPSS 22.00 for Windows. Descriptive statistical analysis was expressed as means and standard deviation (SD) for quantitative variables. Comparative analysis was performed by comparing the mean deviation between planned and performed endodontic access cavities using Student’s *t*-test, since variables had normal distribution; *p* < 0.05 was considered statistically significant.

## 3. Results

The means, SD values for deviation, and statistical significance are displayed in Table 1. Mean horizontal deviations observed at the coronal entry point of endodontic access cavities of the SN were higher in absolute value than those of the DN study group and MN control group (Figure 3).

The paired *t*-test revealed a statistically significant difference between the coronal entry point deviations of SN and MN (Table 1). There was also a statistically significant difference between DN and MN (Table 1). However, there was no statistically significant difference between SN and DN (Table 1) (Figure 3A–F). Mean horizontal deviations observed at the apical endpoint of endodontic access cavities of the SN study group were higher in absolute value than those of the DN study group and MN control group (Figure 3). The paired *t*-test revealed a statistically significant difference between the apical deviations of SN and MN (Table 1). There was also a statistically significant difference between DN and MN (Table 1). However, there was no statistically significant difference between SN and DN (Table 1) (Figure 3A–F). Mean angular deviations observed in the endodontic access cavities of the MN control group were higher than those of the SN and DN study groups (Figure 3). The paired *t*-test revealed a statistically significant difference between the angular deviations of SN and MN (Table 1). There was also a statistically significant difference between DN and MN (Table 1). However, there was no statistically significant difference between SN and DN (Table 1) (Figure 3A–F).

In summary, the DN study group showed more accurate endodontic access cavities than SN study group at the deviation angle and the horizontal deviation (measured at the coronal entry point and the apical endpoint) in absolute value.

All endodontic access cavities performed by the computer-aided navigation system allowed locating the root canal system; however, those performed by the MN control group showed one root perforation and two missed root canals.

## 4. Discussion

The results obtained in the present study accepted the null hypothesis (H0), which is that there would be no difference between the accuracy of endodontic access cavities performed through static and dynamic navigation systems compared with the conventional technique. To date, computer-aided dynamic navigation has been applied in the field of dental implant surgery [1,2,3,4,5]. This technique has improved the accuracy of dental implant placement and reduced clinical complications, making the technique more favorable and predictable [1,2,3,5]. In this study, the computer-aided dynamic navigation system was applied to test this technique in the field of endodontics for the first time to perform endodontic access cavities.

Previously, computer-aided static navigation techniques with surgical templates were developed in an attempt to improve the accuracy of dental implant placement. These techniques require surgical planning based on preoperative CBCT scans and 3D surface scans and allow a better treatment planning and a major comprehension of the anatomical features of the case [21,22]. Datasets obtained from the digital workflow are uploaded to 3D implant planning software to design virtual templates by matching the 3D surface scan and CBCT data by aligning key points of the crown of the teeth [1]. It has been reported that the static guidance technique had a 0.621° mean angle deviation, 0.193 mm mean coronal deviation, and 0.277 mm mean apical deviation in vitro compared to freehand dental implant placement [1]. This technique has been applied in endodontic treatment to improve the conservative access cavities [6,7]. Zehnder et al. (2016) obtained a 1.81° mean angle deviation, 0.16–0.21 mm mean coronal deviation, and 0.17–0.47 mm mean apical deviation in endodontic access cavities performed with a 1.5 mm diameter implant bur [8]. The inaccuracy associated with the static guidance technique could influence the locations of root canals and cause tooth fragility or root perforation. Connert et al. (2019) [9] showed evidence of the existence of unlocated canals in 8.3% of cases and a mean substance loss of 9.8 mm [10]. In this in vitro study, computer-aided navigation study groups allow to locate all root canals planned without complications; however, MN control group showed one root perforation and two missed root canals. The narrow root anatomy of lower central incisors and the high angular deviation associated with MN control group might contribute to the root perforation and missed toot canals appearance. The most relevance parameters analyzed in this study are the angular and apical deviations, because the apical deviation influences over the risk of root perforation and missed root canals and is directly related with the angular deviation in cases of depth endodontic access cavities (calcific metamorphosis) because a high angular deviation increases the horizontal apical endpoint deviation. Despite obtaining better results than the conventional freehand technique, this inaccuracy has promoted the potential application of computer-aided dynamic navigation technology to clinically transfer the positions of virtually planned endodontic access cavities. Moreover, one of the main advantages of this technology is the ability to change the access cavity direction in real time. The stereoscopic motion-tracking optical cameras dynamically recognize and triangulate the optical reference markers, guiding the access cavity at the angle, pathway, and depth that were planned preoperatively. Additional advantages include visibility of dental tissues during clinical procedures and preservation of tooth tissue, reducing the risk of iatrogenic damage [11,12].

Computer-aided dynamic navigation is especially useful in cases of dental development malformations, such as dense invaginatus/evaginatus, in which several accurate and conservative access cavities are needed to localize individual root canals [4,5], even for performing a conservative osteotomy and root-end section in endodontic microsurgery [23,24].

Static guidance requires the development of several surgical templates to allow straight access to individual root canals in posterior teeth. This is not the case with computer-aided dynamic navigation, since needed access cavities are planned preoperatively [14,15,16]. A computer-aided static navigation technique performed with surgical templates avoids the need for drilling guidance during surgery [13,14,15,16,17]. Therefore, endodontic cavity access accuracy directly depends on the design and manufacturing process of the surgical template, so if there is any inaccuracy during the fabrication process, it might cause intraoperative complications. On the other hand, computer-aided dynamic navigation systems allow direct view of the surgical field and provide the ability to relocalize the position of the endodontic access cavity. In addition, it is especially useful in cases of limited mouth opening or posterior region treatments [15,16,17]. The main disadvantage of computer-aided dynamic navigation systems is the difficulty of keeping the system display in sight during the procedure [18]. However, augmented reality devices can be used to transfer a virtual image of the system without losing sight of the therapeutic field [13]. The preoperative planning information of the system is displayed on the mounted laptop computer on the mobile unit. As the “target” is displayed on the laptop unit, the operator is looking away from the patient instead of at the tooth. Maintaining the drill entry point angle, pathway, and depth and controlling the handpiece requires a certain level of technical skill, hand-eye coordination, and manual dexterity that involves a learning curve [1,2,3,4,5,11,12].

In this in vitro study, the endodontic access cavities performed by the DN study group were more accurate than SN study group; however, they were not statistically significant. This may be due to the small sample size, the learning curve attributed to computer-aided dynamic navigation systems, and the established endodontic access cavity depth, because the angular deviations associated to the SN study group showed differences close to the statistical significance with regard to the DN study group, so that if the endodontic access cavities were deeper, the horizontal deviations observed at the apical endpoint would be increasing between the SN study group and the DN study group. In addition, it has been proven that the fracture resistance of single-rooted teeth submitted to conservative endodontic access cavities was no statistically significantly different compared to traditional endodontic accesses cavities. Nevertheless, the mean fracture resistance values related to conservative endodontic access cavities was higher in absolute value than mean fracture resistance values related to traditional endodontic accesses cavities. In addition, conservative endodontic access cavities allowed a lower canal wall area untouched by the endodontic rotary instruments with a lower dentin volume removed in absolute values [25] and the cleaning and shaping capability is the main prognosis factor of the root canal treatment [26]. Nevertheless, the above cited ten conservative endodontic access cavities were performed manually, without computer-aided static or dynamic procedures, following the endodontic access cavities design of Clark and Khademi in 2010. So it is difficult to ensure that the conservative endodontic access cavities presented the same size, location, and design to allow the comparation between them.

The teaching objective derived from this study is that manually performed (freehand) endodontic access cavities result in decreased accuracy compared with computer-aided static and dynamic navigation systems. Further research is needed to determine the accuracy and clinical complications of endodontic access cavities performed with new technologies.

## 5. Conclusions

In conclusion, within the limitations of this study, our results show that computer-aided static and dynamic navigation procedures allow more accurate and safe endodontic access to cavities than conventional freehand techniques.

## Figures and Tables

**Figure 1 jcm-09-00129-f001:**
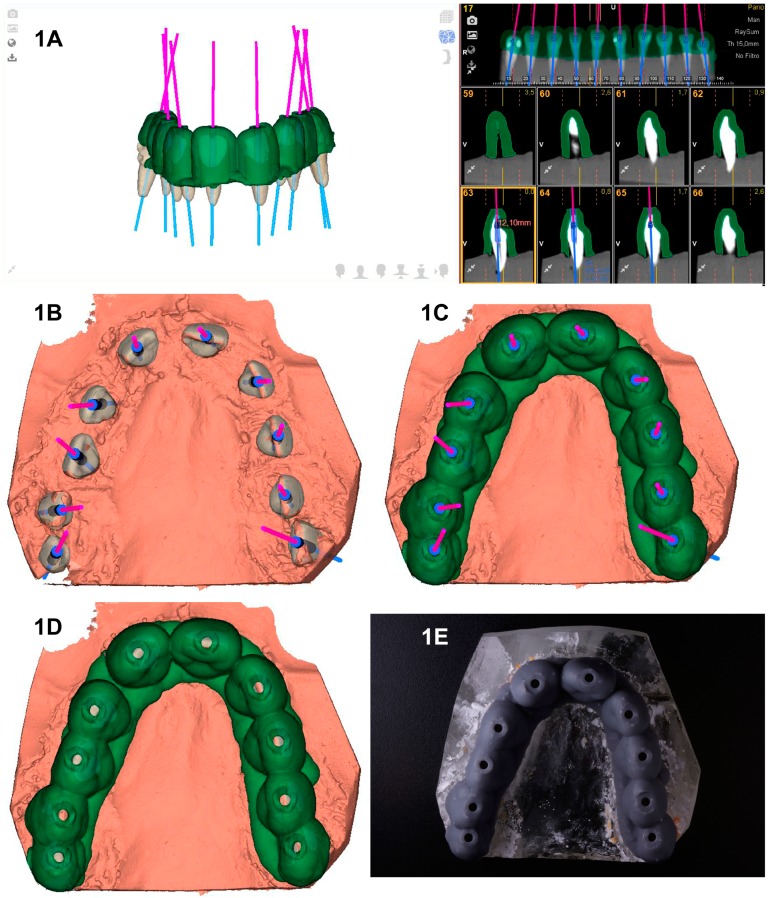
(**A**,**B**) Endodontic access cavities planned with the computer-aided static navigation system incorporating the cone-beam computed tomography (CBCT) scan (pink lines); (**C**,**D**) virtual template design according to planned virtual endodontic access cavities; and (**E**) manufactured stereolithography template fixed over dental surface of teeth placed over epoxy resin model.

**Figure 2 jcm-09-00129-f002:**
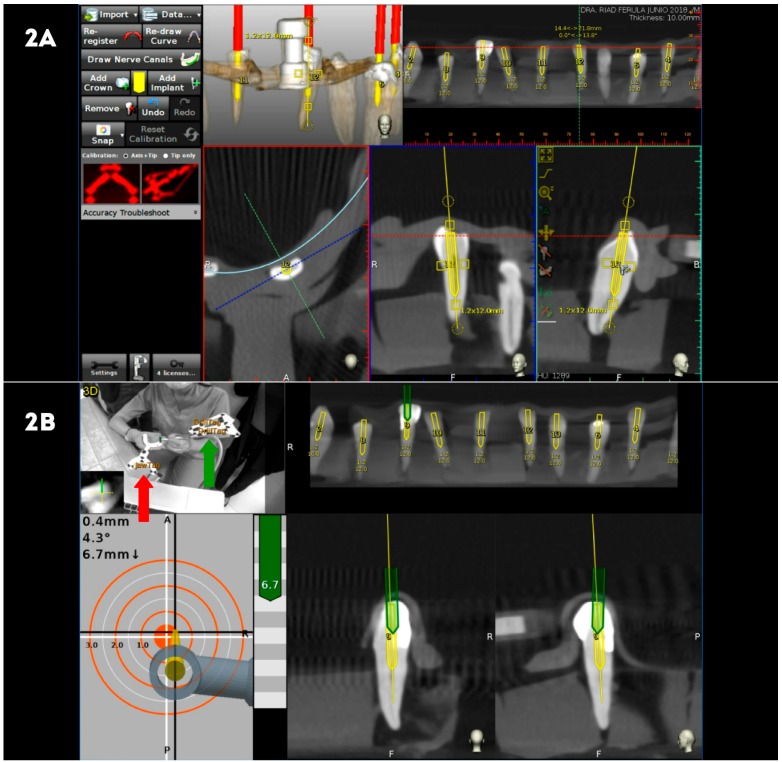
(**A**) Planning endodontic access cavities with treatment planning software of computer-aided dynamic navigation system (yellow cylinders), and (**B**) performing endodontic access cavities (green cylinders) controlled in all planes and depth with the black-and-white jaw tag fixed over the dental surface of the teeth with the thermoplastic template and the black-and-white drill tag (red arrow) attached to the high-speed handpiece (green arrow).

**Figure 3 jcm-09-00129-f003:**
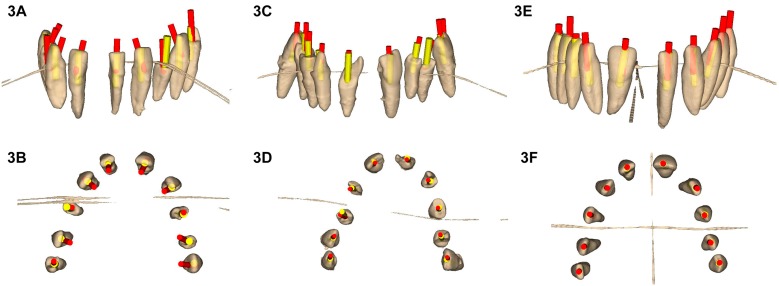
(**A**,**B**) Planned endodontic access cavities (yellow cylinders) and access cavities performed manually (red cylinders). Guided endodontic access cavities planned and performed through (**C**,**D**) computer-aided static navigation system and (**E**,**F**) computer-aided dynamic navigation system.

**Table 1 jcm-09-00129-t001:** Descriptive deviation values at coronal (mm), apical (mm), and angular (°) levels and statistical significance (*p*-value).

		*n*	Mean	SD	Minimum	Maximum	*p*-Value
Coronal	SN	10	7.44	1.57	5.40	10.0	SN-DN = 0.654
	DN	10	3.14	0.86	2.00	5.10	SN-MN < 0.001
	MN	10	4.03	1.93	1.10	7.10	DN-MN < 0.001
Apical	SN	10	7.13	1.73	4.80	9.80	SN-DN = 0.914
	DN	10	2.48	0.94	1.10	3.80	SN-MN < 0.001
	MN	10	2.43	1.23	0.80	4.50	DN-MN < 0.001
Angular	SN	10	10.04	5.20	4.10	19.40	SN-DN = 0.072
	DN	10	5.58	3.23	1.70	10.40	SN-MN < 0.001
	MN	10	14.95	11.15	0.80	29.70	DN-MN < 0.001

SN, static navigation system; DN, dynamic navigation system; MN, manual navigation.

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
