# Peer review of "Accuracy of Computer-Aided Dynamic Navigation Compared to Computer-Aided Static Procedure for Endodontic Access Cavities: An In Vitro Study"

_jcm, 2020, doi:10.3390/jcm9010129_

Round 1

Reviewer 1 Report

Thanks to the authors for the interesting paper.

It could be of interest in the field of endodontics (dentistry) however there are many concerns about the novelty, the clinical significance and the methodology.

Please consider to change in all the text computer-aided static navigation with computer aided static guidance (static id not navigated).

The novelty of this research is limited because other papers already reported the advantages of static guide or dynamic navigation in endodontic cases with difficult access.

Moreover it is not really clinical relevant to create an endodontic access with guide or dynamic navigation considering that teeth with ultraconservative access are not statistically more resistant to fracture than the ones with conservative endodontic access or sometimes (for certain kind of teeth - single rooted included) than the ones with traditional endodontic access. 

Specific comments:

Introduction:

the authors stated "the null hypothesis (H0) stating that there would be no difference between the accuracy of the two techniques", however in the results they stated "there was no statistically significant difference between SN and DN". However, in the discussion they wrote "The results obtained in the present study reject the null hypothesis (H0), which is that there 169 would be no difference between the accuracy of endodontic access cavities performed through static 170 and dynamic navigation systems compared with conventional technique."

Considering the results the null hypothesis should be accepted because the 2 techniques are static and dynamic. Free hand is just the control group. Please revise this through the text.

Table and results section are a repetition. Please use more the table, insert in the table the significativi and do not duplicate the data of the table in the text of the results section.

Authors stated that "Deviations were evaluated in axial, sagittal, and coronal views" however it is not possible to understand in which plan there were the deviations. This is an important parameter in order to understand where is the highest error of each methodology.

Were the 3 groups homogeneous regarding the dimensions of the teeth.

Authors stated that they used "A diamond bur with a diameter of 1.2 mm on the active part, a total 96 length of 14 mm, and a working length of 11 mm was used". However in the figures different numbers are visible... what was the exact numbers used in the computer project?

Did the authors performed a study of the power of the study. Are we sure that 10 teeth/group are enough for a kind of study like this?

Discussion should be improved. The methodology should be more accurate and statistics required some attention.

Probably the study of the accuracy of these 2 systems could be more helpful performing the study on  molars, were the preservation of sound dentine could improve in some (limited) cases their fracture resistance. 

English require some attention.

Author Response

Dear reviewer,

I’m pleased to resubmit the manuscript of the work entitled, “Accuracy of computer-aided dynamic navigation compared to computer-aided static navigation for endodontic access cavities: An in vitro study”.

Reviewer 1: Introduction: the authors stated "the null hypothesis (H0) stating that there would be no difference between the accuracy of the two techniques", however in the results they stated "there was no statistically significant difference between SN and DN". However, in the discussion they wrote "The results obtained in the present study reject the null hypothesis (H0), which is that there 169 would be no difference between the accuracy of endodontic access cavities performed through static 170 and dynamic navigation systems compared with conventional technique.”

Response: In order to adapt to the reviewer's 1 comments, we have rephrasing the paragraph.

Reviewer 1: Table and results section are a repetition. Please use more the table, insert in the table the significativi and do not duplicate the data of the table in the text of the “Results” section

Response: In order to adapt to the reviewer's 1 comments, we have added the statistical significance in the “Table 1” and we have removed the data of the table from the “Results” section.

Reviewer 1: Authors stated that "Deviations were evaluated in axial, sagittal, and coronal views" however it is not possible to understand in which plan there were the deviations. This is an important parameter in order to understand where is the highest error of each methodology.

Response: The therapeutic planning software (NemoScan®, Nemotec, Madrid, Spain) used in the measurement procedure of horizontal (coronal and apical) and angular deviations makes a three-dimensional measurement between the planning and the endodontic access cavities performed, but does not allow individualizing the deviations in each of the planes. It provides a single numerical value, but it is not possible to interpret which plane corresponds to.

Reviewer 1: Were the 3 groups homogeneous regarding the dimensions of the teeth.

Response: In order to adapt to the reviewer's 1 comments, we have we have added a selection criteria in the “Study design” section.

Reviewer 1: Authors stated that they used "A diamond bur with a diameter of 1.2 mm on the active part, a total 96 length of 14 mm, and a working length of 11 mm was used". However in the figures different numbers are visible... what was the exact numbers used in the computer project?

Response: The diameter of the bur was 1.2mm, we regret the mistake made by including an image corresponding to a prior study. The Figure 2 has been changed.

Reviewer 1: Did the authors performed a study of the power of the study. Are we sure that 10 teeth/group are enough for a kind of study like this?

Response: It was thought as a pilot study to analyze the possibility of developing more projects related to this line of research if the results supported it.

Reviewer 1: Discussion should be improved. The methodology should be more accurate and statistics required some attention.

Response: In order to adapt to the reviewer's 1 comments, we have improved the Discussion section with the most actual references.

Reviewer 1: Probably the study of the accuracy of these 2 systems could be more helpful performing the study on molars, were the preservation of sound dentine could improve in some (limited) cases their fracture resistance.

Response: We appreciate the suggestion recommended by the reviewer and will consider it for a future study with a larger sample size.

Reviewer 1: English require some attention.

Response: In order to adapt to the reviewer's 1 comments, we have send the manuscript to a specialized traductor. We attached the Certificate.

We take this opportunity to thank the recommendations and suggestions made by the reviewer to improve the document.

Yours sincerely.

Reviewer 2 Report

This manuscript is an in vitro study of accuracy of CAD dynamic and static navigation for endodontic access cavities. The scope of this manuscript is suitable for journal of clinical medicine.

Materials and methods

There are lack of the description regarding on the intraoperative procedure. For SN group, line 83-98 described the preoperative procedure of the fabrication for guided template.

For DN group, line104-106 described the thermoplastic template, however, it is hard to understand totally with this brief explanation. Providing the figure for this thermoplastic template with a radiographic marker, and black/white jaw tag-handle, will help to understand. For figure 2B, the first figure, with the man drilling the access cavity, showed the red arrow with box. This might need more detailed explanation. As my understanding, the thermoplastic template was inserted and real-time guided access cavity preparation was performed. This process also needs to be provided.

Line 162: conventional method_ is it MN group? It is recommended to discuss regarding on the two missed canal and a perforation cases.

Table1: Result of Statistical analysis should be indicated in the table to identify the statistical difference easily. The schematic diagram for description of those parameters needs to be provided.

In this study, there are three parameter including coronal, apical and angular differences between planned and performed access opening.

According to your manuscript, The “Coronal” is the horizontal discrepancy. The “apical” is apical endpoint. It is hard to understand. Maybe the template upper and lower aperture is presented as those terminology?

Those parameters and its clinical relevance, and results regarding on this data should be discussed in the discussion section.

Author Response

Dear reviewer,

I’m pleased to resubmit the manuscript of the work entitled, “Accuracy of computer-aided dynamic navigation compared to computer-aided static navigation for endodontic access cavities: An in vitro study”.

Reviewer 2: There is lack of the description regarding on the intraoperative procedure. For SN group, line 83-98 described the preoperative procedure of the fabrication for guided template. For DN group, line104-106 described the thermoplastic template, however, it is hard to understand totally with this brief explanation. Providing the figure for this thermoplastic template with a radiographic marker, and black/white jaw tag-handle, will help to understand. For figure 2B, the first figure, with the man drilling the access cavity, showed the red arrow with box. This might need more detailed explanation. As my understanding, the thermoplastic template was inserted and real-time guided access cavity preparation was performed. This process also needs to be provided.

Response: In order to adapt to the reviewer's 2 comments, we have changed the Figure 2 and added a paragraph.

Reviewer 2: Line 162: conventional method_ is it MN group? It is recommended to discuss regarding on the two missed canal and a perforation cases.

Response: In order to adapt to the reviewer's 2 comments, we have replace “conventional method” for “MN study group”, and we have added a paragraph in the “Discussion” section explaining this results.

Reviewer 2: Table1: Result of Statistical analysis should be indicated in the table to identify the statistical difference easily. The schematic diagram for description of those parameters needs to be provided.

Response: In order to adapt to the reviewer's 2 comments, we have added the statistical significance in the “Table 1”.

Reviewer 2: In this study, there are three parameter including coronal, apical and angular differences between planned and performed access opening. According to your manuscript, The “Coronal” is the horizontal discrepancy. The “apical” is apical endpoint. It is hard to understand. Maybe the template upper and lower aperture is presented as those terminology? Those parameters and its clinical relevance, and results regarding on this data should be discussed in the discussion section.

Response: In order to adapt to the reviewer's 2 comments, we have added a paragraph in the “Measurement procedure” section of the “Material and Methods” section and another one on the “Discussion” section.

Reviewer 2: English language and style are fine/minor spell check required.

Response: In order to adapt to the reviewer's 2 comments, we have sent the manuscript to a specialized translator who has reviewed the style and punctuation. We attached the Certificate.

We take this opportunity to thank the recommendations and suggestions made by the reviewer to improve the document.

Yours sincerely.

Reviewer 3 Report

The computer aided navigation can improve the results of endodontic treatment, since endodontics are considered blinded procedures.

The article is peculiar and represents a novelty for a more accurate therapeutic procedures.

Besides, you did not cited one recent article on navident for endodontic surgery, which demonstrate accuracy in retrieve small anatomical features and which can effort your thesis.

Gambarini G, Galli M, Stefanelli LV, Di Nardo D, Morese A, Seracchiani M, De
Angelis F, Di Carlo S, Testarelli L. Endodontic Microsurgery Using Dynamic
Navigation System: A Case Report. J Endod. 2019 Nov;45(11):1397-1402.e6.

You can also effort the useful administration of CBCT in endodontics by citing the following articles:

Gambarini G, Piasecki L, Ropini P, Miccoli G, Nardo DD, Testarelli L.
Cone-beam computed tomographic analysis on root and canal morphology of
mandibular first permanent molar among multiracial population in Western European population. Eur J Dent. 2018 Jul-Sep;12(3):434-438.

Gambarini G, Piasecki L, Miccoli G, Gaimari G, Nardo DD, Testarelli L.
Cone-beam computed tomography in the assessment of periapical lesions in
endodontically treated teeth. Eur J Dent. 2018 Jan-Mar;12(1):136-143.

You can also add this very novel paper for endodontic purposes that will
effort the significance of the paper:

Hawkins TK, Wealleans JA, Pratt AM, Ray JJ. Targeted endodontic
microsurgery and endodontic microsurgery: a surgical simulation
comparison. Int Endod J. 2019 Nov 1. doi: 10.1111/iej.13243. [Epub ahead
of print] PubMed PMID: 31674678.

Author Response

Dear reviewer,

I’m pleased to resubmit the manuscript of the work entitled, “Accuracy of computer-aided dynamic navigation compared to computer-aided static navigation for endodontic access cavities: An in vitro study”.

Reviewer 3: English language and style are fine/minor spell check required.

Response: In order to adapt to the reviewer's 3 comments, we have sent the manuscript to a specialized translator who has reviewed the style and punctuation. We attached the Certificate.

Reviewer 3: Besides, you did not cited one recent article on navident for endodontic surgery, which demonstrate accuracy in retrieve small anatomical features and which can effort your thesis. Gambarini G, Galli M, Stefanelli LV, Di Nardo D, Morese A, Seracchiani M, De Angelis F, Di Carlo S, Testarelli L. Endodontic Microsurgery Using Dynamic Navigation System: A Case Report. J Endod. 2019 Nov;45(11):1397-1402.e6.

Response: In order to adapt to the reviewer's 3 comments, we have added a paragraph in the “Discussion” section and the “Reference”.

Reviewer 3: You can also effort the useful administration of CBCT in endodontics by citing the following articles: Gambarini G, Piasecki L, Ropini P, Miccoli G, Nardo DD, Testarelli L. Cone-beam computed tomographic analysis on root and canal morphology of mandibular first permanent molar among multiracial population in Western European population. Eur J Dent. 2018 Jul-Sep;12(3):434-438.

Gambarini G, Piasecki L, Miccoli G, Gaimari G, Nardo DD, Testarelli L. Cone-beam computed tomography in the assessment of periapical lesions in endodontically treated teeth. Eur J Dent. 2018 Jan-Mar;12(1):136-143.

Response: In order to adapt to the reviewer's 3 comments, we have added a paragraph in the “Discussion” section and the “Reference”.

Reviewer 3: You can also add this very novel paper for endodontic purposes that will effort the significance of the paper: Hawkins TK, Wealleans JA, Pratt AM, Ray JJ. Targeted endodontic microsurgery and endodontic microsurgery: a surgical simulation comparison. Int Endod J. 2019 Nov 1. doi: 10.1111/iej.13243. [Epub ahead of print] PubMed PMID: 31674678.

Response: In order to adapt to the reviewer's 3 comments, we have added the “Reference”.

We take this opportunity to thank the recommendations and suggestions made by the reviewers to improve the document.

Yours sincerely,

Round 2

Reviewer 1 Report

Thanks to the author to revised the paper.

Unfortunately some of my concerns remain the same.

Title was not changed and again there is static navigation that there are 2 words in contrast.

I suggest to add "in absolute value", after higher, in the results section were the author stated "Mean horizontal deviations observed at the coronal entry point of endodontic access cavities of the SN were higher than those of the DN study group and MN control group".

In general add "in absolute value" every time you declare a difference but it is not statically significant. I understood you would like to point the attention on the higher precision of DN than SN, however, if there is not significance, authors should avoid to use higher for DN than SN or at least they should specify "in absolute value".

Nothing is higher or lower in the results section of a scientific paper if the significance of statistic is not reached.

Thanks to add the phrase about the homogeneity of the teeth... but are the author sure about the phrase and the homogeneity now stated?... In the figure 2 it seems that the teeth are not so homogenous.

The improvement of the discussion section is really poor.

Please discuss the limits of the study; the absence of significance between DN and SN; the absence of clinical relevance to use this sophisticated approaches just for a contracted access cavity especially in single rooted teeth were it is already well demonstrated that there is not difference in the resistance of incisors with traditional or conservative access cavity

"Impacts of Conservative Endodontic Cavity on Root Canal Instrumentation Efficacy and Resistance to Fracture Assessed in Incisors, Premolars, and Molars

R Krishan et al. J Endod 40 (8), 1160-6. Aug 2014. PMID 25069925.";   Please discuss the limits of the methodology used, standardisation, pilot study and so on.        

Author Response

Dear reviewer,

I’m pleased to resubmit the manuscript of the work entitled, “Accuracy of computer-aided dynamic navigation compared to computer-aided static procedure for endodontic access cavities: An in vitro study”.

Reviewer 1: Title was not changed and again there is static navigation that there are 2 words in contrast.

Response: In order to adapt to the reviewer's 1 comments, we have we have changed the title.

Reviewer 1: I suggest to add "in absolute value", after higher, in the results section were the author stated "Mean horizontal deviations observed at the coronal entry point of endodontic access cavities of the SN were higher than those of the DN study group and MN control group".

Response: In order to adapt to the reviewer's 1 comments, we have we have added this sentence in the “Results” and “Discussion” sections.

Reviewer 1: In general add "in absolute value" every time you declare a difference but it is not statically significant. I understood you would like to point the attention on the higher precision of DN than SN, however, if there is not significance, authors should avoid to use higher for DN than SN or at least they should specify "in absolute value".

Response: In order to adapt to the reviewer's 1 comments, we have added the sentence “in absolute value” where correspond and we have written the following sentence to highlight the higher precision of DN study group than SN study group: “In summary, the DN study group showed more accurate endodontic access cavities than SN study group at the deviation angle and the horizontal deviation (measured at the coronal entry point and the apical endpoint) in absolute value”.

Reviewer 1: Nothing is higher or lower in the results section of a scientific paper if the significance of statistic is not reached.

Response: In order to adapt to the reviewer's 1 comments, we have we have added the sentence “in absolute value” after higher, as the reviewer said earlier: “I suggest to add "in absolute value", after higher, in the results section were the author stated "Mean horizontal deviations observed at the coronal entry point of endodontic access cavities of the SN were higher than those of the DN study group and MN control”.

Reviewer 1: Thanks to add the phrase about the homogeneity of the teeth... but are the author sure about the phrase and the homogeneity now stated?... In the figure 2 it seems that the teeth are not so homogenous.

Response: In order to respond to the reviewer's 1 comments, we clarify that the selected teeth were measured with an electronic calliper and presented clinical crown dimensions between 8.5-9.5mm high, 4.5-5.5mm wide M-D and 5.5-6.5mm wide V-L / P. Those who did not comply with these measures were discarded. Perhaps small variations in the radiographic projection at the time of the capture of Figure 2 (the teeth were placed in the form of a dental arch, as can be seen in Figures 1 and 3) can suggest that they did not have the predetermined dimensions.

Reviewer 1: The improvement of the discussion section is really poor.

Response: In order to adapt to the reviewer's 1 comments, we have we have improve the “Discussion” section developing the questions asked by the reviewer

Reviewer 1: Please discuss the limits of the study; the absence of significance between DN and SN; the absence of clinical relevance to use this sophisticated approaches just for a contracted access cavity especially in single rooted teeth were it is already well demonstrated that there is not difference in the resistance of incisors with traditional or conservative access cavity

Response: In order to adapt to the reviewer's 1 comments, we have we have added a sentence in the “Discussion” sentence.

Reviewer 1: "Impacts of Conservative Endodontic Cavity on Root Canal Instrumentation Efficacy and Resistance to Fracture Assessed in Incisors, Premolars, and Molars. R Krishan et al. J Endod 40 (8), 1160-6. Aug 2014. PMID 25069925.";   Please discuss the limits of the methodology used, standardisation, pilot study and so on.

Response: In order to adapt to the reviewer's 1 comments, we have we have added a sentence in the “Discussion” sentence.

We take this opportunity to thank the recommendations and suggestions made by the reviewer to improve the document.